# Effect of Soybean Protein Isolate-7s on Delphinidin-3-*O*-Glucoside from Purple Corn Stability and Their Interactional Characterization

**DOI:** 10.3390/foods11070895

**Published:** 2022-03-22

**Authors:** Dongxia Chen, Yuheng Liu, Jia Li, Xiaozhen Sun, Jiadong Gu, Yang He, Hui Ci, Liankui Wen, Hansong Yu, Xiuying Xu

**Affiliations:** 1College of Food Science and Engineering, Jilin Agricultural University, Changchun 130118, China; chendongxia1218@163.com (D.C.); lyhh5689@163.com (Y.L.); lijia9918@163.com (J.L.); sunxiaozhen825@163.com (X.S.); gu14709526592@163.com (J.G.); heyang200704@jlau.edu.cn (Y.H.); wenliankui@jlau.edu.cn (L.W.); yuhansong@jlau.edu.cn (H.Y.); 2Food Inspection Department, National Ratafia and Fruits and Vegetables Drinks Quality Surveillance Test Center, Tonghua 134000, China; tonghuazhijian@163.com; 3Division of Soybean Processing, Soybean Research & Development Center, Chinese Agricultural Research System, Changchun 130118, China; 4National Engineering Laboratory for Wheatand Corn Deep Processing, Changchun 130118, China

**Keywords:** anthocyanins, protein, co-pigmentation, light-thermal stabilities

## Abstract

Anthocyanins are abundant in purple corn and beneficial to human health. Soybean protein isolate-7s (SPI-7s) could enhance the stability of anthocyanins. The stable system of soybean protein isolate-7s and delphinidin-3-*O*-glucoside complex (SPI-7s-D3G) was optimized using the Box–Behnken design at pH 2.8 and pH 6.8. Under the condition of pH 2.8, SPI-7s effectively improved the sunlight-thermal stabilities of delphinidin-3-*O*-glucoside (D3G). The thermal degradation of D3G conformed to the first order kinetics within 100 min, the negative enthalpy value and positive entropy value indicated that interaction was caused by electrostatic interaction, and the negative Gibbs free energy value reflected a spontaneous interaction between SPI-7s and D3G. The interaction of SPI-7s-D3G was evaluated by ultraviolet visible spectroscopy, circular dichroism spectroscopy and fluorescence spectroscopy. The results showed that the maximum absorption peak was redshifted with increasing the *α*-helix content and decreasing the *β*-sheet contents, and D3G quenched the intrinsic fluorescence of SPI-7s by static quenching. There was one binding site in the SPI-7s and D3G stable system. The secondary structure of SPI-7s had changed and the complex was more stable. The stabilized SPI-7s-D3G will have broad application prospects in functional foods.

## 1. Introduction

Anthocyanins are nature water-soluble pigments that belong to a class of flavonoids [1]. They benefit human health with their anti-cancer, anti-oxidative, anti-inflammatory, and vision protecting properties [2]. Anthocyanins are widely found in fruits and various plants, such as blueberry, strawberry, bilberry, and purple corn [3,4,5]. Purple corn (*Zea Mays* L.) is a rich source of anthocyanins [6]. D3G is the main monomer of the purple corn, which is very sensitive to environmental factors such as temperature, light, pH, oxygen, ascorbic acid, enzymes, and metal ions [7]. Food processing (especially thermal processing) and storage are two important factors in the production and distribution of food, which can easily result in significant loss and degradation of anthocyanins [8]. Suitable stabilization methods can promote the stability of anthocyanin products. Stable anthocyanins can also be used as a natural pigment in beverages, cakes, candies and convenience foods to provide bright colors, or they can be added to functional foods as functional ingredients [1,9]. Therefore, enhancing the stability of anthocyanins is necessary [10,11].

At present, several stabilization methods have been developed to improve the stability of anthocyanins such as microencapsulation (spray/freeze-drying) [12] and glycosyl acylation [13]. In addition, phenolic, polysaccharides, and protein can be added to improve the stability of anthocyanins [14,15,16]. The emulsification of protein can enhance the stability of anthocyanins [17]. Specifically, Qin et al. [18] report that over the pH 2–7 range the whey protein isolate (WPI) can improve the stability of anthocyanins. Similarly, Zang et al. [19] also reported that the interaction between WPI and blueberry anthocyanins was caused by hydrophobic forces via hydrogen bonds using ultraviolet visible spectroscopy (UV-Vis) and fluorescence spectroscopy. The interaction between protein and anthocyanins could improve the binding affinity by changing the secondary structure of protein, thus effectively protecting the stability of anthocyanins [20].

Soybean is one of the most economical, nutritious, and important food crops in legumes [21]. SPI is the most common soy protein product, which is derived from soybean meal and contains 90% Dry Weight protein [22]. Compared with animal protein, SPI has attracted extensive attention due to its higher nutritional value, lower cost, and richer functions [23]. Chen et al. [24] found that preheated SPI improved the thermal stability and oxidation stability of cyanidin-3-*O*-glucoside (C3G) because the C3G can form complexes with SPI through hydrophobic interactions. SPI is a mixture of various proteins, and its main components are divided into four categories according to its sedimentation coefficient 2s, 7s, 11s and 15s [25]. SPI-7s has good processing characteristics, such as solubility, gelation, water and oil retention and emulsifying ability. At present, the mechanism of improving anthocyanins stability by SPI-7s is rarely studied. The stabilized SPI-7s-D3G will have potentially good application value in functional food.

This study aimed to construct the stable system of SPI-7s-D3G and investigate the interaction mechanism of SPI-7s and D3G through UV-Vis, circular dichroism spectroscopy (CD) and fluorescence spectroscopy. This will promote the wide application of anthocyanins in the food industry.

## 2. Materials and Methods

### 2.1. Materials

SPI was purchased from Shanghai Yuanye Biotechnology Co., Ltd. (Shanghai, China). Fresh purple corn was purchased from the local market (Changchun, China) in August 2020, and the cob and kernel were dried at 50 °C. All other chemicals used were of analytical grade.

### 2.2. Extraction of SPI-7s

The SPI-7s was extracted according to the method of Liu et al. [26] with slight modifications. SPI was dispersed in deionized water (1:15 *w*/*v*) and the pH was adjusted to 7.5 using NaOH (2N) then centrifuged (LXJ-IIB, Shanghai Anting Scientific Instrument Factory, Shanghai, China) at 9000× *g* force for 30 min and retained the supernatant. The sodium bisulfite (0.98 g/L) was added to supernatant and its pH was adjusted to 6.4 with HCl (2N), followed by centrifugation at 6500× *g* for 20 min at 4 °C. The supernatant was adjusted to pH 5.0 after 1h stirring (MS-H, Bona technology, Shenzhen, China), and then centrifuged at 9000× *g* for 30 min at 4 °C. The result of precipitate was SPI-7s fraction, the pH was adjusted to 7.5 and then freeze-dried (LG 0.2, Xinyang quick-frozen equipment manufacturing Co., Ltd., Xinyang, China). The protein content of obtained SPI-7s was 95.21%, as determined by the Kjeldahl method [26].

### 2.3. Extraction of D3G

The anthocyanins of purple corn were extracted according to the method of He et al. [27] with slight modifications. The dried cob and kernel were extracted by solution (1% HCl-65% ethanol, *v*/*v*) in water bath at 60 °C in the dark. The supernatant was concentrated to a density of (1.0–1.1) × 10^3^ kg/m^3^. The product was purified with D101 macroporous resin, and then eluted using ethanol, concentrated and freeze dried until it needed to be analyzed (anthocyanins content ≥ 25%, moisture content ≤ 10%). The D3G monomer was prepared and collected with preparative HPLC (Waters 2545, Milford, MA, USA) according to the following conditions [28]: The injection volume was 500 μL with UV detection at 530 nm. The separation of purple corn anthocyanins was performed by using a SunFire^®^ Prep C18 OBDTM (19 mm × 150 mm, 5 μm, Waters, Milford, MA, USA) at a flow rate of 5 mL/min and column temperature was 25 °C. The eluents used consist of methyl alcohol (A) and 5% formic acid-water solution (B). The gradient elution program was initiated from: 15% eluent A for 0–5 min; 15–20% eluent A for 5–8 min; 20–25% eluent A for 8–13 min; 25–31% eluent A for 13–16 min; 31–100% eluent A for 16–17 min; 100% eluent A for 17–19 min; 100–15% eluent A for 19–22 min; 15% eluent A for 22–30 min; 15–100% eluent A for 30–33 min. The purity of the D3G monomer was ≥95%, the relevant HPLC chromatogram was shown in Appendix A.

### 2.4. Optimum Design of SPI-7s-D3G Stable System

Complex proportion between SPI-7s and D3G (the system was dissolved in pH 2.8/pH 6.8 PBS, final concentration of D3G was 20 mg/100mL) (A, 20:1, 30:1, 40:1 *w*/*w*), reaction temperature (B, 50, 60, 70 °C), and reaction time (C, 60, 90, 120 min) were selected using Box–Behnken design. The absorbance at 521 nm was measured and the co-pigmentation rate (C) was calculated using the following Equation (1).

(1)
$$C\;\left(\%\right)=\left[\left(A-A_{0}\right)/A_{0}\right]\times 100$$

where A is the absorbance of SPI-7s-D3G complex, and A_0_ is the absorbance of D3G.

### 2.5. Stability of SPI-7s-D3G

#### 2.5.1. Light Stability

The stability test of SPI-7s-D3G by sunlight treatments was designed according to previous studies [29] with slight modifications. The sunlight illumination of SPI-7s-D3G (pH 2.8/pH 6.8) was studied under sunlight (samples were placed on the indoor windowsill and the daily insolation time was not less than 6 h) at room temperature (20 ± 2 °C) on 0, 7, 14, 21, and 28 days. The untreated D3G was used as control, and the retention rate for sunlight stability analysis is shown in Equation (2).

(2)
$$R\;\left(\%\right)=\left(A_{t}/A\right)\times 100$$

where A is the absorbance before sunlight or thermal stability treatment, and A_t_ is the absorbance after sunlight or thermal stability treatment.

#### 2.5.2. Thermal Stability

The thermal stability test was carried out according to previous studies [18] with slight modifications. The thermal stability of SPI-7s-D3G (pH 2.8/pH 6.8) was evaluated in the water bath at 100 °C at 0.5, 1, 1.5, and 4 h. The untreated D3G was used as control, and the retention rate for thermal stability analysis is shown in Equation (2).

Degradation kinetics was carried out according to the method of Attaribo et al. [30], with slight modifications. The samples (pH 2.8/pH 6.8) were prepared in tin foil with final sample concentrations of 30 mg/mL SPI-7s and 1 mg/mL D3G. The sample solutions were heated in a water bath at 100 °C for 100 min. After every 20 min, the samples were cooled in ice quickly. The untreated D3G was used as the control. Two dilutions of each sample were prepared, one with potassium chloride buffer (pH 1.0) and the other with sodium acetate buffer (pH 4.5). The measurement TAC is shown in Equation (3).

(3)
$$\mathrm{TAC}\;(\%)=\frac{\left[{\left(A_{521}\;{-\;A}_{700}\right)}_{\mathrm{pH}\;1.0}\;-\;{\left(A_{521}\;{-\;A}_{700}\right)}_{\mathrm{pH}\;4.5}\right]\;\times \;M\;\times \;\mathrm{DF}\;\times \;V}{{\varepsilon \;\times \;W}_{t}}\;\times \;1000$$

where A_521_ and A_700_ are the absorbance different of the samples at λ_521_ nm and λ_700_ nm, M is molecular weight of D3G (465.3 mol/L), DF is the dilution factor of the samples during measurement, V is the volume of samples (mL), ε is the molar extinction coefficient of D3G (29,600 mol/L·cm), and W_t_ is the weight of samples (mg).

The degradation rate (k) and half-life (t_1/2_) of the samples during heat treatment are shown in Equations (4) and (5):ln (C/C_0_) = −kt(4)
t_1/2_ = −ln(0.5)/k(5)
where C_0_ and C are the anthocyanins concentration before and after heating, t is the time of heating.

The enthalpy (ΔH) and entropy (ΔS) can be calculated by the Va not Hoff Equation (6). The value of Gibbs free energy (ΔG) can be acquired by the following Equation (7).
lnK_s_ = −ΔH/RT + ΔS/R(6)
ΔG = ΔH − TΔS(7)
where R is the gas constant (8.314 J·mol^−1^·K^−1^), T is the temperature at 298 K, 308 K or 318 K.

### 2.6. Characterization of SPI-7s-D3G

#### 2.6.1. UV-Vis

The UV-Vis absorption spectra of D3G in the absence and presence of SPI-7s in PBS (pH 2.8/pH 6.8) were measured using a T6 UV-Vis spectrophotometer (Beijing Universal Instrument Co., Ltd., Beijing, China) over a wavelength range of 400–600 nm.

#### 2.6.2. CD Spectroscopy

CD spectroscopy was carried out using a MOS-450 spectrometer (Biologic Company, Seyssinet-Pariset, France) in the far-UV region (200–250 nm). The SPI-7s in the absence and presence of D3G in PBS (pH 2.8/pH 6.8) were measured in a quartz cuvette with a path length of 0.1 cm. The scanning rate, response rate, spectral resolution, and bandwidth were set at 15 nm/min, 0.2 nm, 0.25 s, and 0.5 nm, respectively.

#### 2.6.3. Fluorescence Spectroscopy

The fluorescence spectra of the SPI-7s and SPI-7s-D3G were recorded using a Fluoromax-4 fluorophotometer (HORIBA Company, American, Irvine, CA, USA). Samples were prepared by mixing a fixed concentration of SPI-7s in PBS (pH 2.8/pH 6.8) with different concentrations of D3G. The final concentrations of D3G in the mixtures were 10, 20, 30, 40 and 50 μM. Fluorescence spectra were recorded at an excitation wavelength of 280 nm and an emission wavelength of 300–400 nm at 298 K, 308 K or 318 K. The excitation and emission slit widths were 5 nm.

To further understand the fluorescence quenching mechanism, the fluorescence date can be calculated in the following Stern–Volmer Equation (8):F_0_/F = 1 + K_sv_Cq = 1 + K_q_t_0_C_q_(8)
where F and F_0_ are the fluorescence intensity in the absence and presence of quencher, respectively, K_sv_ and K_q_ are the quenching constant and quenching rate constant, C_q_ is the concentration of D3G, and t_0_ is the average lifetime of the biomolecule, which is around 10^−8^ s [31].

The binding constant (K_s_) and binding site number (*n*) were calculated from the double logarithmic Stern–Volmer Equation (9):
log [(F_0_ − F)/F] = logK_s_ + *n*logC_q_(9)

### 2.7. Statistical Analysis

All measurements were performed in triplicate and results were expressed as mean standard deviation (±SD). The ANOVA test was performed using OriginLab and Design-Expert 8.0.6. *p* value < 0.05 or < 0.01 were considered as significantly different.

## 3. Results and Discussion

### 3.1. Optimum Design of SPI-7s-D3G Stable System

The Box–Behnken software was used for fitting analysis, the following Y_1_ (pH 2.8) equation model of absorbance was obtained:Y_1_ = 0.77 + 0.024A + 0.026B + 0.007750C − 0.011AB − 0.0015AC + 0.013BC − 0.054A^2^ − 0.031B^2^ − 0.027C^2^

Y_2_ (pH 6.8) equation model of absorbance was obtained:Y_2_ = 0.72 + 0.018A + 0.028B − 0.0015C − 0.015AB − 0.00325AC + 0.00475BC − 0.031A^2^ − 0.029B^2^ − 0.018C^2^

Table 1 and Appendix A show that the model and regression were significant (*p* < 0.01). The effect sequence of each factor on the absorbance was B (reaction temperature) > A (complex proportion between SPI-7s and D3G) > C (reaction time). The results show that the model had a well-fitting degree and could be used to direct the process of SPI-7s and D3G complex.

The Box–Behnken software was used to solve the regression equation, and the optimum technology of the SPI-7s-D3G (pH 2.8/pH 6.8) was concluded as follows: complex proportion was 32.56:1/31.70:1 (*w*/*w*), reaction temperature was 63.70/64.42 °C, and reaction time was 62.17/67.43 min, theoretical absorbance was 0.739/0.777, respectively. According to the actual situation, the pH 2.8/pH 6.8 stable system construction parameters were jointly adjusted: complex proportion was 30:1 (*w*/*w*), reaction temperature was 60 °C, and reaction time was 60 min, and their absorbance were 0.711 and 0.723, respectively. The absorbance and the co-pigmentation rate of SPI-7s-D3G were increased to 51.21%/39.57% and 42.62%/40.70%, respectively. The results show that SPI-7s can improve D3G co-pigmentation effect, and SPI-7s-D3G has high co-pigmentation effect of at pH 2.8.

As stated above, SPI-7s can improve the co-pigmentation effect of D3G by heating (60 °C) for 60 min. At present, the interaction between protein and anthocyanins is usually carried out at room temperature and ordinary pressure. As Attaribo et al. [30] reported, the silkworm pupae protein was heated in a water bath at 80 °C for 30 min, and then stirred with C3G at 25 °C for 1 h; the protein significantly increased the co-pigmentation effect of anthocyanins. WPI preheated at 80 °C for 30 min has been reported to significantly improve the stability of C3G by complexation when they are stirred at 25 °C for 30 min [29]. At present, protein needs to be preheated in advance before mixing with anthocyanins, but generally the preheating temperature is too high, or the preheating time is too long. In the future, protein and anthocyanins can be mixed first and then heated, which can not only shorten the reaction time, but also lower the reaction temperature and improve the economic benefits. According to the reported, He et al. [27] used the high hydrostatic pressure-assisted (300 MPa) technique in order to shorten the reaction time between the organic acids and anthocyanins, and then enhanced the stability of anthocyanins. In the future, techniques such as the high hydrostatic pressure-assisted are expected to facilitate protein–anthocyanins interactions.

### 3.2. Stability of SPI-7s-D3G

#### 3.2.1. Light Stability

Figure 1 and Figure 2 show the absorbance and retention rate of SPI-7s-D3G and D3G when they were treated with sunlight (28 d, 20 ± 2 °C) in PBS (pH 2.8/pH 6.8). The absorbance and retention rate of SPI-7s-D3G were increased to 41.02%/39.23% and 16.94%/14.85%, respectively. A similar report by He et al. [29] indicates the retention rate of WPI-C3G was increased to 9.2%, when subjected to photo illumination treatment at pH 3.6 (10 d). The retention rate of SPI-7s-D3G and WPI-C3G were improved, which indicated that protein could enhance the stability of anthocyanins under light treatments. As shown in the results, the retention rate of SPI-7s-D3G had a higher value in the acidic system recorded compared to the neutral system.

#### 3.2.2. Thermal Stability

Figure 3 and Figure 4 show the absorbance and retention rate of D3G when they were subjected to heat treatment (2 h) in the absence and presence of SPI-7s in PBS (pH 2.8/pH 6.8). The absorbance and retention rate of SPI-7s-D3G were increased to 40.34%/37.95% and 17.33%/14.93%, respectively. These results indicate that SPI-7s improved the stability of D3G more efficiently in acidic condition than neutral condition. A similar study showed that the retention rate of C3G at pH 3.5 increased from 55.00% to 67.50% when compounded with rice glutelin fibrils [32]. It might be the higher stability of flavium cation form of anthocyanins in low pH environment [33]. The retention rate of the purple-fleshed sweet potato anthocyanin extracts and soy protein complex was increased by 16.2%, when it was heated by 100 °C for 30 min at pH 3.0 [34]. He et al. [29] also found that the retention rate of WPI-C3G was increased by 9.4% after heating by 80 °C for 8 h at pH 3.6. The results indicate that the interaction between protein and anthocyanins could protect the stability of anthocyanins during heat treatment.

The degradation kinetic parameters of D3G alone or SPI-7s-D3G are as shown in Figure 5 and Table 2. Correlation coefficients (R_2_) of the linear curve were greater than 0.95, indicating that the thermal degradation of D3G conforms to the first order kinetics within 100 min [35]. The k and t_1/2_ values of D3G (pH 2.8/pH 6.8) alone were 0.00185 × 10^−5^/0.0021 × 10^−5^ min^−1^ and 374.67/313.64 min. The k decreased to 0.00128 × 10^−5^/0.00155 × 10^−5^ min^−1^, and t_1/2_ increased to 841.37/736.58 min, respectively, when the SPI-7s interaction with D3G. The results showed that the interaction between SPI-7s and D3G can protect the stability of D3G during heat treatment (Table 2). At pH 2.8, the k was the lowest and the t_1/2_ was the highest, which indicated that SPI-7s-D3G had the best affinity and thermal stability under acidic conditions.

Research showed that the interaction between protein and anthocyanins was dominated by molecular interactions such as hydrophobic interactions, van der Waals forces, electrostatic interactions, and hydrogen bonding, which further improved the stability of anthocyanins [34]. Specifically, when ΔH > and ΔS > 0, hydrophobic interactions played a major role; when ΔH < 0 and ΔS < 0, the interaction forces were mainly hydrogen bond and van der Waals forces; when ΔH > 0 and ΔS < 0, they were mainly electrostatic and hydrophobic interactions; when ΔH < 0 and Δ S > 0, the electrostatic interactions were the main interactions [35]. According to Table 3, ΔH < 0 and ΔS > 0 during the reaction of SPI-7s and D3G (pH 2.8/pH 6.8) indicated that the main forces acting between SPI-7s and D3G were electrostatic interactions. Furthermore, ΔG < 0, indicated that the binding between SPI-7s and D3G was spontaneous. Lang et al. [36] reported that the main forces acting during the reaction of bovine serum albumin and malvidin-3-*O*-galactoside were electrostatic interactions. However, Zang et al. [19] proved that the hydrophobic interactions dominated between WPI and malvidin-3-*O*-galactoside. Thus, the interaction force between protein and anthocyanins depends on their structural characteristics.

At present, there is a theoretical basis for studying the interaction between protein and anthocyanins. We only studied the stability of the interaction between protein and anthocyanins under sunlight and heat treatments. The food processing such as high temperature, high pressure, high salt, high sugar, irradiation and microwave can cause a large loss of anthocyanins, thus limiting anthocyanins application in food [37]. According to a relevant report, bovine serum albumin and WPI is able to protect the antioxidant activity of blueberry anthocyanins under the treatment of photo illumination, vitamin C and sucrose [19,38]. However, the stability of anthocyanins under extreme conditions such as high temperature, high pressure and radiation needs to be studied, which is of great significance to the development of anthocyanins functional food.

### 3.3. Characterization of SPI-7s-D3G

#### 3.3.1. UV-Vis Spectra

The UV-Vis spectra of SPI-7s-D3G and D3G (pH 2.8/pH 6.8) were recorded in Figure 6. The results showed that when D3G combined with SPI-7s, the intensity of D3G changed and the absorption peaks increased by 4/2 nm. This indicated that SPI-7s and D3G produced intermolecular or intramolecular co-pigmentation, which produced a redshift effect on the maximum absorption wavelength of D3G in a visible range [27]. Meanwhile, SPI-7s-D3G had high stability in the acidic system, because the absorption peak intensity of pH 2.8 was higher than pH 6.8.

#### 3.3.2. CD Spectra

CD spectroscopy is a sensitive method for studying the secondary structures of protein [32]. In the secondary structure of protein, it is generally considered that the *α*-helix is the most stable structure, *β*-Folding is a better stable structure, and random helix is the most unstable structure [26,33]. Figure 7A,B showed the characteristic absorption peak of SPI-7s-D3G (pH 2.8/pH 6.8) at 208 nm. When D3G was added to the SPI-7s, the characteristic absorption peak shifted to 209/210 nm, respectively. The negative ellipticity was decreased, resulting in changes in the *α*-helix, *β*-sheet, *β*-turn, and random coil contents of the SPI-7s [30]. The contents of the *α*-helix, *β*-sheet, *β*-turn, and random coil of SPI-7s and SPI-7s-D3G were calculated using the CDPro software, as presented in Table 3. When SPI-7s mixed with D3G, the *α*-helix content of SPI-7s was increased by 15.4%/13.67%, the *β*-sheet content of SPI-7s was decreased by 11.7%/9.34%, and the random coil content of SPI-7s was decreased by 4.85%/5.64%, respectively. According to a previous report [24], mixing SPI with D3G could increase the *α*-helix content of SPI and decrease the *β*-sheet content, thereby improving the stability of anthocyanins. Similar studies showed that *α*-helix content increases and the *β*-sheet content decreases for bovine serum albumin bonded with malvidin-3-*O*-galactoside [36].

#### 3.3.3. Fluorescence Spectra

Fluorescence spectroscopy is widely used to research the structural changes in protein in complex material, such as food products [37]. The protein emission spectra of intrinsic fluorescence were determined by the tryptophan (Try, regard Try as the dominant factor) and tyrosine (Tyr), when the excitation wavelength is 280 nm. Changes in the physical and chemical environment in the protein can result in a change in emission peaks [39]. Figure 7C,D shows the fluorescence spectra of SPI-7s and SPI-7s-D3G in the PBS (pH 2.8/pH 6.8). By increasing the D3G concentration, the fluorescence intensity of SPI-7s decreased, which indicated that there was an interaction between the SPI-7s and D3G [37]. In addition, the maximum emission wavelengths of SPI-7s were redshifted by 4/5 nm, respectively. The results showed that the addition of D3G changed the microenvironment of SPI-7s, rearranged its Trp and Tyr residues and increased its polarity, which may be the reason for the change in protein secondary structure [22]. Similar studies showed that when adding C3G, the polarity of microenvironment around the preheating silkworm pupae protein was enhanced, and the maximum emission wavelength of the complex was redshifted [30].

Fluorescence quenching of small molecules on the surface of protein may occur in the processes of reaction, energy transfer, complex formation, and collision quenching [40]. The mechanism of protein fluorescence quenching can be divided into static quenching and dynamic quenching; when the K_sv_ increases with temperature, it is dynamic quenching, whereas it is static quenching on the contrary. According to the data in Figure 8 and Figure 9 and Table 4, the Stern–Volmer equation curves presented a good linear relationship at pH 2.8 and pH 6.8, indicating that the quenching of SPI-7s only occurred through one quenching mechanism. In order to clarify the fluorescence quenching mechanism between SPI-7s and D3G, the relationship between F_0_/F and D3G concentration was plotted [37]. The fluorescence quenching was static quenching because the K_sv_ values of SPI-7s-D3G decreased with rising temperature. As shown in the double logarithm regression equation, the *n* values were approximately equal to 1, which suggested only one binding site was in both SPI-7s and D3G during their interactions, the same as reported by Meng [41].

These results indicated that the biding between SPI-7s and D3G was spontaneous through electrostatic interaction, which increased the content of α-helix and decreased the content of β-sheet. The secondary structure of protein was changed by static quenching, and a stable complex was formed, which was more obvious under acidic conditions. The research showed that the stabilizing effect of the interaction between different proteins and different anthocyanins was different, which may be affected by the source of protein and the structure of anthocyanins. For example, the quenching mechanism between WPI and purple carrot anthocyanin occurred dynamically through hydrogen bonding [42], while the quenching mechanism between WPI and blueberry anthocyanin was static quenching by hydrophobicity [19]. The different interactions between anthocyanins and protein were attributed to the structural diversity of anthocyanins. [43]. On the other hand, the interaction between the same anthocyanins and different proteins was different. As Fu et al. reported [33], the results of molecular docking showed that hydrogen bonding and van der Waals forces were both the acting force in the binding of egg ovalbumin and C3G. CD spectroscopy analysis showed that α-helix content increased and β-sheet content decreased and the remaining content of C3G is higher than 90% after 2 h of heating treatment at 80 °C. Additionally, Chen et al. reported [24] the complex between SPI and C3G was mainly due to hydrophobicity. CD spectroscopy analysis showed that α-helix content decreased, and β-sheet content increased, and the degradation rate of C3G decreased by 70% after 1 h of treatment at 80 °C. This may be due to the different amino acid composition of proteins, which made the binding sites of proteins and anthocyanins were different, and protein improved the stability of anthocyanins to varying degrees. In the future, systematic research and analysis on the stable structure between protein and anthocyanins will be beneficial to the market expansion.

## 4. Conclusions

The stable system of SPI-7s-D3G (pH 2.8/pH 6.8) was optimized using the Box–Behnken design, and the co-pigmentation rate of SPI-7s-D3G was increased to 42.62%/40.70%, respectively. The retention rate of SPI-7s-D3G was increased to 16.94%/14.85%, 17.33%/14.93% when it was treated with sunlight treatment at 20 ± 2 °C for 28 d or heating at 100 °C for 2 h, respectively. This means the SPI-7s can protect the sunlight-thermal stabilities. The thermal degradation of D3G conforms to the first order kinetics within 100 min. When the SPI-7s interacted with D3G, the k decreased to 0.00128 × 10^−5^/0.00155 × 10^−5^ min^−1^ and t_1/2_ increased to 841.37/736.58 min. The negative enthalpy (ΔH) value (−23.22/−21.95 kJ·mol^−1^) and positive entropy (ΔS) value (27.93/30.59 kJ·mol^−1^) indicated that the interaction was caused by electrostatic interaction, and the negative Gibbs free energy (ΔG) value (−23.22/−21.98 kJ·mol^−1^) reflected a spontaneous interaction between SPI-7s and D3G. The ultraviolet visible spectroscopy analysis showed that the maximum absorption peak was redshifted by 2/4 nm. Circular dichroism spectroscopy analysis showed that the content of *α*-helix was increased by 13.67/15.40%, the content of *β*-sheet was decreased by 9.34/11.70% and the random coil content of SPI-7s was decreased by 4.85%/5.64%, which indicated that SPI-7s and D3G formed a complex with a more stable structure. Fluorescence spectroscopy analysis showed that the fluorescence intensity of SPI-7s decreased, which indicated that there was an interaction between the SPI-7s and D3G. D3G quenched the intrinsic fluorescence of SPI-7s by static quenching, and the secondary structure of the SPI-7s had changed. There was one binding site in SPI-7s and D3G binding systems. After the interaction between SPI-7s and D3G, the secondary structure of SPI-7s had changed, which made the complex more stable. Considering the potential health benefits of D3G and SPI-7s on the human body, and the ability of SPI-7s to improve the stability of anthocyanin, the stabilized SPI-7s-D3G will have potentially good application value in functional food.

## Figures and Tables

**Figure 1 foods-11-00895-f001:**
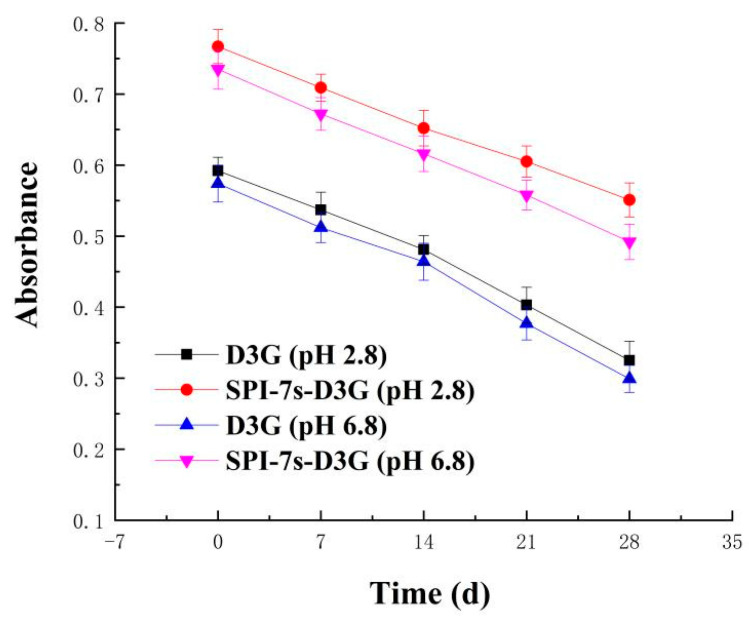
Absorbance value of sunlight treatment at room temperature. SPI-7s-D3G: The complex of soybean protein isolate-7s and delphinidin-3-*O*-glucoside.

**Figure 2 foods-11-00895-f002:**
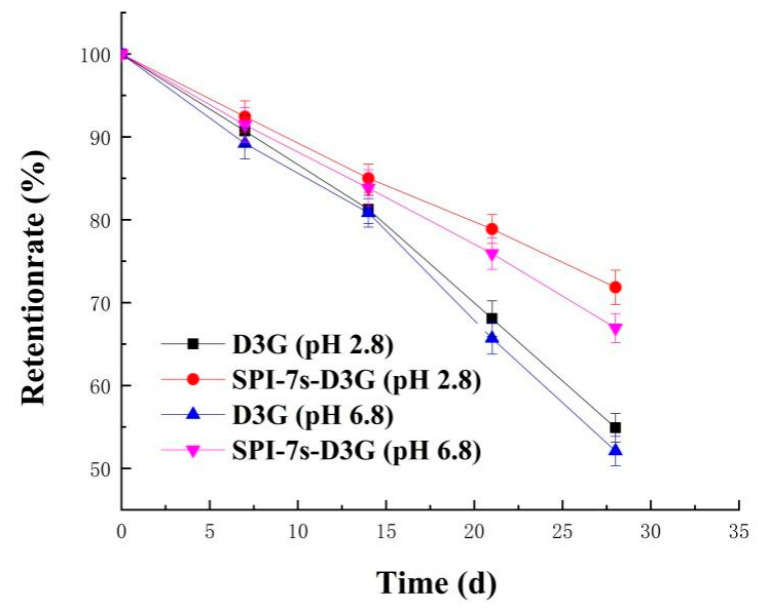
Retention rate of sunlight treatment at room temperature. SPI-7s-D3G: The complex of soybean protein isolate-7s and delphinidin-3-*O*-glucoside.

**Figure 3 foods-11-00895-f003:**
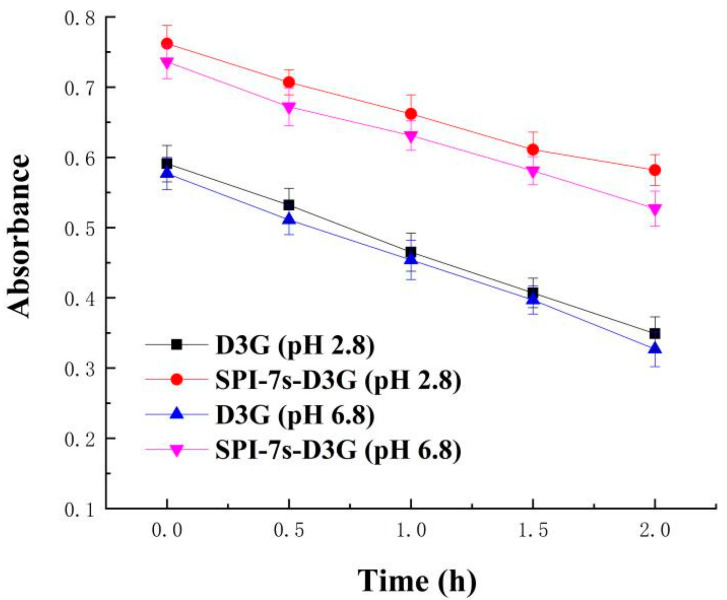
Absorbance value of heat treatment at 100 °C for 2 h. SPI-7s-D3G: The complex of soybean protein isolate-7s and delphinidin-3-*O*-glucoside.

**Figure 4 foods-11-00895-f004:**
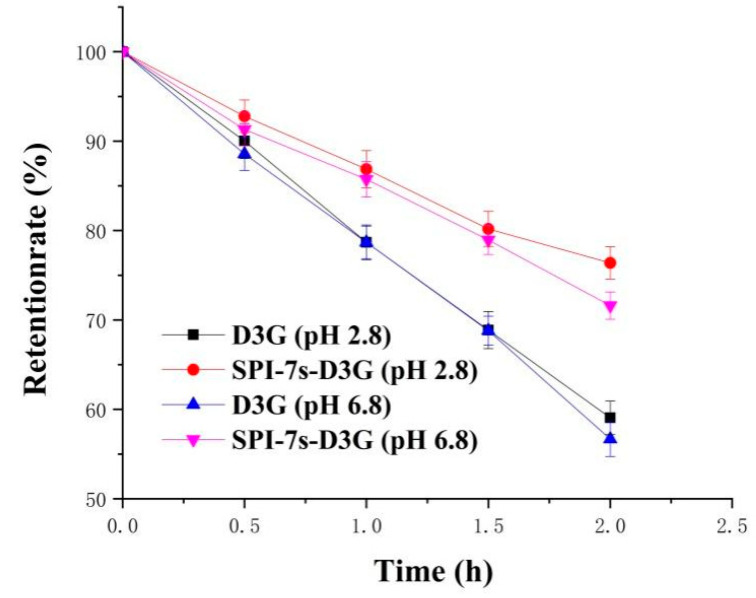
Retention rate of heat treatment at 100 °C for 2 h. SPI-7s-D3G: The complex of soybean protein isolate-7s and delphinidin-3-*O*-glucoside.

**Figure 5 foods-11-00895-f005:**
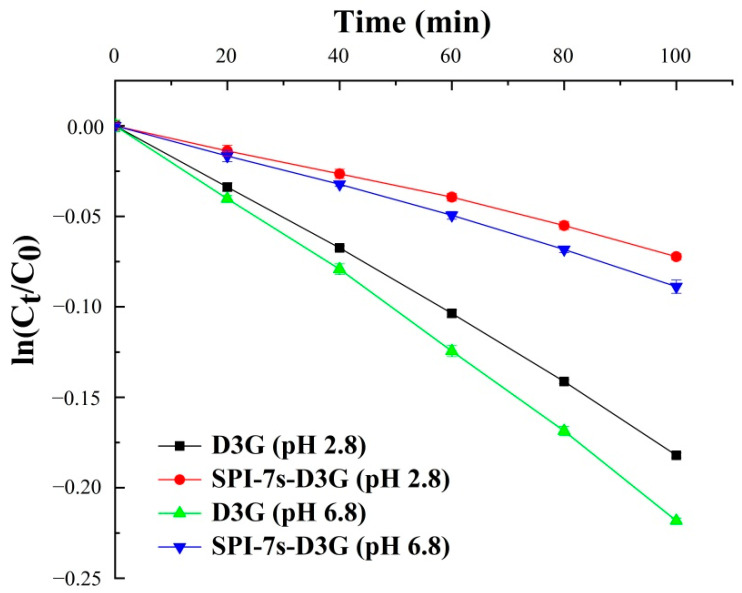
Thermal degradation analysis at 100 °C for 100 min. SPI-7s-D3G: The complex of soybean protein isolate-7s and delphinidin-3-*O*-glucoside.

**Figure 6 foods-11-00895-f006:**
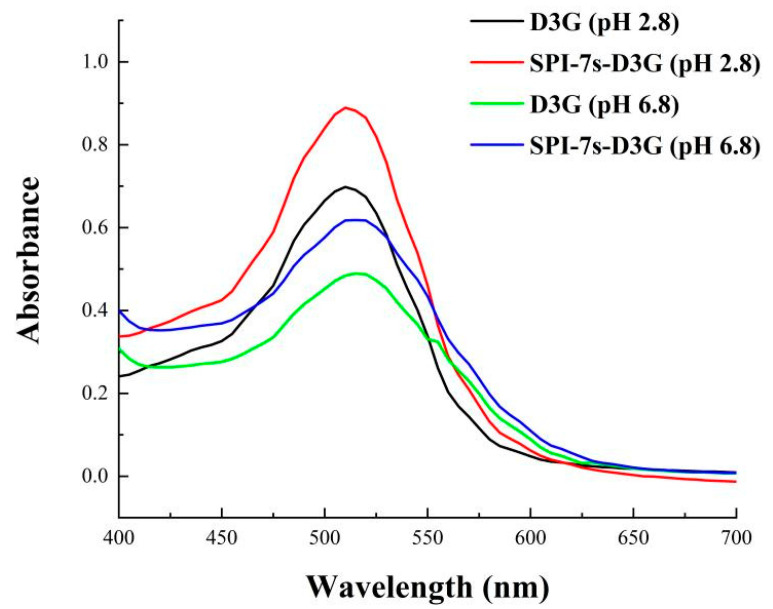
Ultraviolet visible spectra of stable system. SPI-7s-D3G: The complex of soybean protein isolate-7s and delphinidin-3-*O*-glucoside.

**Figure 7 foods-11-00895-f007:**
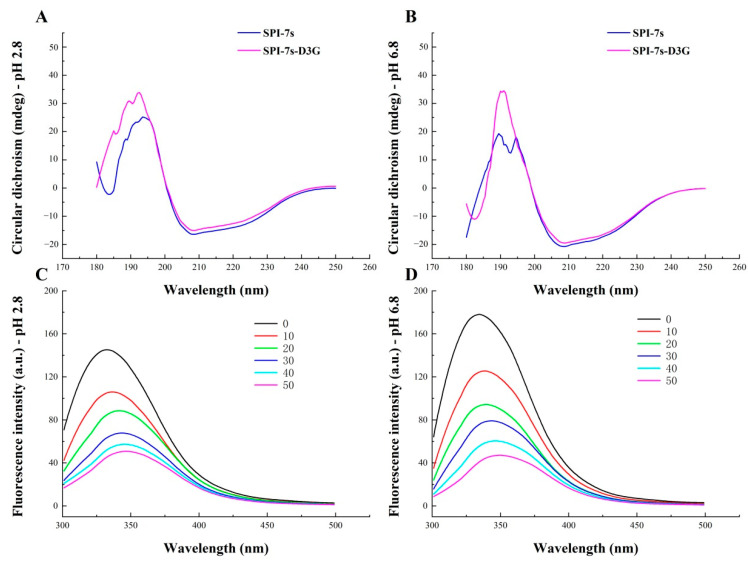
Circular dichroism spectra and Fluorescence spectra of stable system. Circular dichroism spectra of pH 2.8 (**A**), Circular dichroism spectra of pH 6.8 (**B**), Fluorescence spectra of pH 2.8 (**C**), Fluorescence spectra of pH 6.8 (**D**). SPI-7s-D3G: The complex of soybean protein isolate-7s and delphinidin-3-*O*-glucoside; 0–50: D3G concentration (μmol).

**Figure 8 foods-11-00895-f008:**
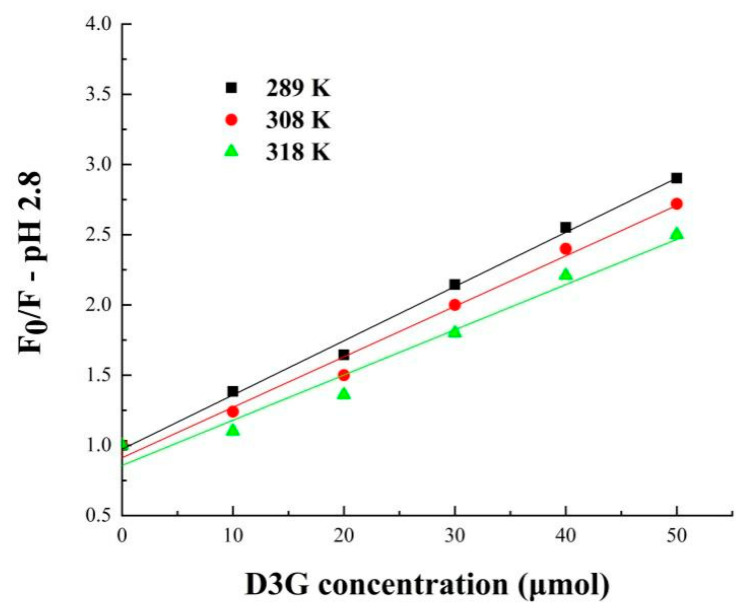
Stern–Volmer plots for SPI-7s-D3G (pH 2.8). SPI-7s-D3G: The complex of soybean protein isolate-7s and delphinidin-3-*O*-glucoside.

**Figure 9 foods-11-00895-f009:**
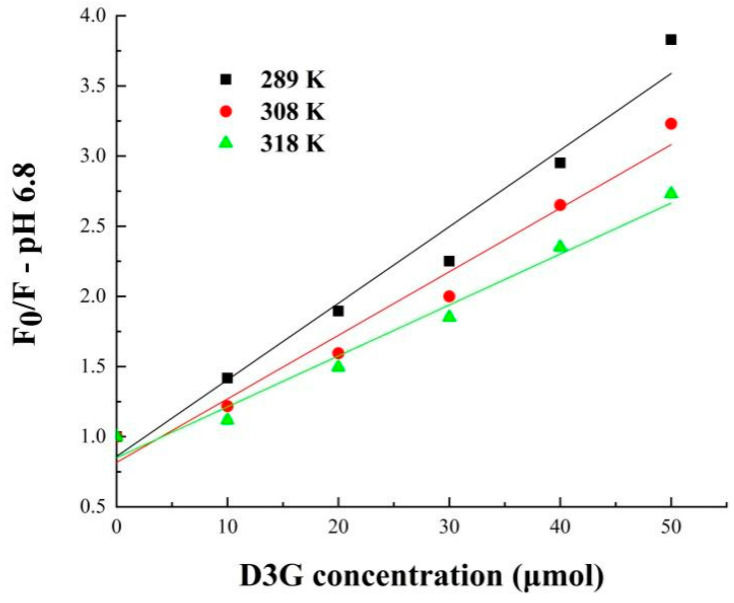
Stern–Volmer plots for SPI-7s-D3G (pH 6.8). SPI-7s-D3G: The complex of soybean protein isolate-7s and delphinidin-3-*O*-glucoside.

**Table 1 foods-11-00895-t001:** Optimum design of SPI-7s-D3G ^†^ stable system (pH 2.8) (mean ± SD) (*n* = 3).

No.	A: Complex Proportion		B: Reaction Temperature (°C)		C: Reaction Time (min)	Abs ^‡^
1	1 (30:1)		0 (60)		0 (60)	0.774
2	1		−1 (50)		−1 (30)	0.689
3	−1 (40:1)		1 (70)		0	0.721
4	0 (20:1)		1		0	0.694
5	−1		0		−1	0.703
6	1		0		0	0.787
7	1		0		0	0.768
8	1		0		0	0.771
9	1		0		0	0.774
10	1		1		1 (90)	0.759
11	0		0		−1	0.655
12	0		0		1	0.677
13	1		1		−1	0.721
14	1		−1		1	0.675
15	−1		−1		0	0.697
16	0		−1		0	0.625
17	−1		0		1	0.719
Source	Sum of square	df	Mean square	*F* value	*p* value	Significant
Model	0.021	9	0.002354	24.24	0.0002	**
A	0.003698	1	0.003698	38.07	0.0005	**
B	0.005202	1	0.005202	53.55	0.0002	**
C	0.000008	1	0.000008	0.082	0.7824	
AB	0.000625	1	0.000625	6.43	0.0389	*
AC	0.000049	1	0.000049	0.50	0.5005	
BC	0.000036	1	0.000036	0.37	0.5619	
A^2^	0.004725	1	0.004725	48.64	0.0002	**
B^2^	0.003917	1	0.003917	40.32	0.0004	**
C^2^	0.001769	1	0.001769	18.22	0.0037	**
Residual	0.00068	7	0.00009714			
Lack of Fit	0.00005	3	0.00001667	0.11	0.9524	Not significant
Pure Error	0.00063	4	0.001575			
Cor Total	0.022	16				
R^2^_adj_		0.9289				
R^2^_Pred_		0.9184				
C.V./%		1.42				

SPI-7s-D3G ^†^: The complex of soybean protein isolate-7s and delphinidin-3-*O*-glucoside; Abs ^‡^: Absorbance; *: The difference was significant (*p* < 0.05); **: The difference was extremely significant (*p* < 0.01).

**Table 2 foods-11-00895-t002:** Reaction kinetics and thermodynamic parameters.

Sample	k (min^−1^)		R^2^	t_1/2_ (min)
D3G (pH 2.8)	0.00185 ± 3.53 × 10^−5^		0.998	374.67 ± 7.15
SPI-7s-D3G ^†^ (pH 2.8)	0.00057 ± 1.46 × 10^−5^		0.997	1216.04 ± 31.17
D3G (pH 6.8)	0.00221 ± 3.64 × 10^−5^		0.998	313.64 ± 5.17
SPI-7s-D3G (pH 6.8)	0.00066 ± 1.69 × 10^−5^		0.997	1050.22 ± 26.91
Sample	T (K)	ΔH (kJ·mol^−1^)	ΔG (kJ·mol^−1^)	ΔS (J·mol^−1^·K^−1^)
SPI-7s-D3G (pH 2.8)	298	−23.22	−31.54	27.93
	308		−31.82	
	318		−32.10	
SPI-7s-D3G (pH 6.8)	298	−21.98	−31.10	30.59
	308		−30.40	
	318		−31.71	

SPI-7s-D3G ^†^: The complex of soybean protein isolate-7s and delphinidin-3-*O*-glucoside; k: Degradation reaction rate constant; R^2^: Correlation coefficients; t_1/2_: Half-lives; T: The temperature at 298 K, 308 K or 318 K; ΔH: Enthalpy change; ΔG: Free energy change; ΔS: Entropy change.

**Table 3 foods-11-00895-t003:** Circular dichroism spectra.

Sample	*α*-Helix (%)	*β*-Sheets (%)	*β*-Turn (%)	Random Coil (%)
SPI-7s-D3G ^†^ (pH 2.8)	38.13	20.78	21.02	20.07
SPI-7s (pH 2.8)	22.73	32.48	19.87	24.92
SPI-7s-D3G (pH 6.8)	38.03	23.64	20.96	17.37
SPI-7s (pH 6.8)	24.36	32.98	19.65	23.01

SPI-7s-D3G ^†^: The complex of soybean protein isolate-7s and delphinidin-3-*O*-glucoside.

**Table 4 foods-11-00895-t004:** The quenching constants, binding constant and site numbers of SPI-7s-D3G.

Sample	T (K)	K_sv_ (×10^3^ L·mol^−1^)	K_q_ (×10^11^ L·mol^−1^·s^−1^)	*n*
SPI-7s-D3G ^†^ (pH 2.8)	298	38.62	38.62	1.02
	308	35.94	35.94	0.95
	318	32.20	32.20	0.95
SPI-7s-D3G (pH 6.8)	298	54.58	54.58	1.15
	308	45.30	45.30	1.16
	318	36.30	36.30	1.19

SPI-7s-D3G ^†^: The complex of soybean protein isolate-7s and delphinidin-3-*O*-glucoside; T: The temperature at 298 K, 308 K or 318 K; K_sv_: Quenching constant; K_q_: Quenching rate constant; *n*: Number of binding bits.

## Data Availability

Data is contained within the article.

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
