# Peer review of "Effect of Soybean Protein Isolate-7s on Delphinidin-3-O-Glucoside from Purple Corn Stability and Their Interactional Characterization"

_foods, 2022, doi:10.3390/foods11070895_

Round 1
Reviewer 1 Report
The article titled" Effect of soybean protein isolate-7s on delphinidin-3-O-glucoside from purple corn stability and their interactional characterization" could be potentially interesting for Foods readers but it still needs major revision.
Comments and recommendations of the Reviewer are listed below:
- lines 19, 36 - explanation of abbreviation D3G appears for the first time in line 17, so it is no necessary to define it for the second time.
- line 53 - it should be: "using ultraviolet visible spectroscopy"
- There is no aim of the study. Please define the aim at the end of introduction.
- line 56 - please give an appropriate literature to the statement.
- lines 91-92 - the content of protein was determined by Kjeldahl method - there is no literature data - please complete.
- line 94 - it should be: "were extracted".
- line 97 - the density is not clear, please explain, why such value was chosen.
- lines 216-217 - the statement is not clear, please rearrange it.
- line 225 - it should be: "needs"
- titles of figures 3-9 are not clear/correct, please rewrite them.
- lines 264-265 - the style of the sentence is not correct, please rewrite it.
- lines 301-302 - the sentence is not correct and needs to be rewritten.
- lines 417-420 - the sentence is not clear, what do the Authors mean by: "position of hydroxyl and methoxy group in glycine"? - please explain.
- lines 441-442 - the style of the sentence should be corrected.
- lines 456-458 - style of the sentence is not correct (benefits on human body?)
- line 105 - please explain what do the Authors mean by: "of aqueous"
Author Response
Response to Reviewer 1 Comments
Point 1: lines 19, 36 - explanation of abbreviation D3G appears for the first time in line 17, so it is no necessary to define it for the second time.
Response 1: Thanks for your correction. Line 19 “ SPI-7s effectively improved the sunlight-thermal stabilities of delphinidin-3-O-glucoside (D3G).” has been changed to “SPI-7s effectively improved the sunlight-thermal stabilities of D3G.” And Line 36 “Delphinidin-3-O-glucoside (D3G) is the main monomer of the purple corn, which is very sensitive to the environmental factors such as temperature, light, pH, oxygen, ascorbic acid, enzymes, and metal ions ” has been changed to “D3G is the main monomer of the purple corn, which is very sensitive to the environmental factors such as temperature, light, pH, oxygen, ascorbic acid, enzymes, and metal ions”
Point 2: line 53 - it should be: "using ultraviolet visible spectroscopy".
Response 2: Line 53 “Similarly, Zang et al. [19] also reported that the interaction between WPI and blueberry anthocyanins was caused by hydrophobic forces and via hydrogen bonds using by ultraviolet visible spectroscopy (UV-Vis) and fluorescence spectroscopy. ” has been changed to “Similarly, Zang et al. [19] also reported that the interaction between WPI and blueberry anthocyanins was caused by hydrophobic forces and via hydrogen bonds using ultraviolet visible spectroscopy (UV-Vis) and fluorescence spectroscopy. ”
Point 3: There is no aim of the study. Please define the aim at the end of introduction.
Response 3: Line 72-75 “The stable system of soybean protein isolate-7s-delphinidin-3-O-glucoside (SPI-7s-D3G) was constructed in this study at pH 2.8 and pH 6.8, respectively. The co-pigmentation rate, light stability, and thermal stability of SPI-7s-D3G were investigated and then the interaction mechanism of SPI-7s-D3G was studied by ultraviolet visible spectroscopy, circular dichroism spectroscopy, fluorescence spectroscopy. This study will promote the wide application of anthocyanins in food industry.” has been changed to “This study aimed to construct the stable system of SPI-7s-D3G, and investigate the interaction mechanism of SPI-7s and D3G through UV-Vis, circular dichroism spectroscopy (CD), fluorescence spectroscopy. This will promote the wide application of anthocyanins in food industry.”
Point 4: line 56 - please give an appropriate literature to the statement.
Response 4: Line 56 “The interaction between protein and anthocyanins could improve the binding affinity by changing the secondary structure of protein, thus effectively protecting the stability of anthocyanins [20]” we added references here “He, Z.; Xu, M.; Zeng, M.; Qin, F.; Chen, J. Preheated milk proteins improve the stability of grape skin anthocyanins extracts. Food Chemistry 2016, 210, 221-227.”
Point 5: lines 91-92 - the content of protein was determined by Kjeldahl method - there is no literature data - please complete.
Response 5: Line 91-92 “The protein content of obtained SPI-7s was 95.21% determined by Kjeldahl method [26]” we added references here “Liu, C.; Wang, H.; Cui, Z.; He, X.; Wang, X.; Zeng, X.; Ma, H. Optimization of extraction and isolation for 11s and 7s globulins of soybean seed storage protein. Food Chemistry 2007, 102, 1310-1316.”
Point 6: line 94 - it should be: "were extracted".
Response 6: Line 94 “The anthocyanins of purple corn was extracted according to the method of He et al. [26] with slight modifications. ” has been changed to “The anthocyanins of purple corn were extracted according to the method of He et al. [27] with slight modifications. ”
Point 7: line 97 - the density is not clear, please explain, why such value was chosen.
Response 7: The supernatant was concentrated to a density of (1.0−1.1) ×103 kg/m3 which guaranteed its suitable fluidity for further purification.(Line 98)
Point 8: lines 216-217 - the statement is not clear, please rearrange it.
Response 8: Line 216-217 “The parameters were adjusted according to the actual condition: complex proportion was 30:1/30:1 (w/w), reaction temperature was 60/60 ℃, and reaction time was 60/60 min, the absorbance was 0.711/0.723, respectively.” has been changed to “According to the actual situation, the pH 2.8/pH 6.8 stable system construction parameters are jointly adjusted to: complex proportion was 30:1 (w/w), reaction temperature was 60 ℃, and reaction time was 60 min, and their absorbance were 0.711 and 0.723, respectively.”
Point 9: line 225 - it should be: "needs"
Response 9: Line 225 “At present, protein need to be preheated in advance before mixing with anthocyanins, but generally the preheating temperature is too high or the preheating time is too long.” has been changed to “At present, protein needs to be preheated in advance before mixing with anthocyanins, but generally the preheating temperature is too high or the preheating time is too long.”
Point 10: titles of figures 3-9 are not clear/correct, please rewrite them.
Response 10: We modified the titles of figures 3-7. “Figure 3. Absorbance value of heat treatment. SPI-7s-D3G: The complex of soybean protein isolate-7s and delphinidin-3-O-glucoside; pH 2.8: The systems of pH 2.8; pH 6.8: The systems of pH 6.8.” has been changed to “Figure 3. Absorbance value of heat treatment at 100 ℃ for 2 h. SPI-7s-D3G: The complex of soybean protein isolate-7s and delphinidin-3-O-glucoside; pH 2.8: The systems of pH 2.8; pH 6.8: The systems of pH 6.8.”
“Figure 4. Retention rate of heat treatment. SPI-7s-D3G: The complex of soybean protein isolate-7s and delphinidin-3-O-glucoside; pH 2.8: The systems of pH 2.8; pH 6.8: The systems of pH 6.8.” has been change to “Figure 4. Retention rate of heat treatment at 100 ℃ for 2 h. SPI-7s-D3G: The complex of soybean protein isolate-7s and delphinidin-3-O-glucoside; pH 2.8: The systems of pH 2.8; pH 6.8: The systems of pH 6.8.”
“Figure 5. Thermal degradation analysis. SPI-7s-D3G: The complex of soybean protein isolate-7s and delphinidin-3-O-glucoside; pH 2.8: The systems of pH 2.8; pH 6.8: The systems of pH 6.8.” has been changed to “Figure 5. Thermal degradation analysis at 100 ℃ for 100 min. SPI-7s-D3G: The complex of soybean protein isolate-7s and delphinidin-3-O-glucoside; pH 2.8: The systems of pH 2.8; pH 6.8: The systems of pH 6.8.”
“Figure 6. Ultraviolet visible spectra. SPI-7s-D3G: The complex of soybean protein isolate-7s and delphinidin-3-O-glucoside; pH 2.8: The systems of pH 2.8; pH 6.8: The systems of pH 6.8.” has been changed to “Figure 6. Ultraviolet visible spectra of stable system. SPI-7s-D3G: The complex of soybean protein isolate-7s and delphinidin-3-O-glucoside; pH 2.8: The systems of pH 2.8; pH 6.8: The systems of pH 6.8.”
“Figure 7. Circular dichroism spectra and Fluorescence spectra. Circular dichroism spectra of pH 2.8 (A), Circular dichroism spectra of pH 6.8 (B), Fluorescence spectra of pH 2.8 (C), Fluorescence spectra of pH 6.8 (D). SPI-7s-D3G: The complex of soybean protein isolate-7s and delphinidin-3-O-glucoside; pH 2.8: The systems of pH 2.8; pH 6.8: The systems of pH 6.8; 0‒50: D3G concentration (μmol).” has been changed to “Figure 7. Circular dichroism spectra and Fluorescence spectra of stable system. Circular dichroism spectra of pH 2.8 (A), Circular dichroism spectra of pH 6.8 (B), Fluorescence spectra of pH 2.8 (C), Fluorescence spectra of pH 6.8 (D). SPI-7s-D3G: The complex of soybean protein isolate-7s and delphinidin-3-O-glucoside; pH 2.8: The systems of pH 2.8; pH 6.8: The systems of pH 6.8; 0‒50: D3G concentration (μmol).”
“Figure 8. Stern-Volmer plots for SPI-7s-D3G (pH 2.8). SPI-7s-D3G: The complex of soybean protein isolate-7s and delphinidin-3-O-glucoside; pH 2.8: The systems of pH 2.8; pH 6.8: The systems of pH 6.8;289 K,308 K,318 K: The temperature at 298 K, 308 K or 318 K.” has been changed to “Figure 8. Stern-Volmer plots of SPI-7s-D3G (pH 2.8). SPI-7s-D3G: The complex of soybean protein isolate-7s and delphinidin-3-O-glucoside; pH 2.8: The systems of pH 2.8; pH 6.8: The systems of pH 6.8;289 K,308 K,318 K: The temperature at 298 K, 308 K or 318 K.”
“Figure 9. Stern-Volmer plots for SPI-7s-D3G (pH 6.8). SPI-7s-D3G: The complex of soybean protein isolate-7s and delphinidin-3-O-glucoside; pH 2.8: The systems of pH 2.8; pH 6.8: The systems of pH 6.8; 289 K,308 K,318 K: The temperature at 298 K, 308 K or 318 K.” has been changed to “Figure 9. Stern-Volmer plots of SPI-7s-D3G (pH 6.8). SPI-7s-D3G: The complex of soybean protein isolate-7s and delphinidin-3-O-glucoside; pH 2.8: The systems of pH 2.8; pH 6.8: The systems of pH 6.8; 289 K,308 K,318 K: The temperature at 298 K, 308 K or 318 K.”
Point 11: lines 264-265 - the style of the sentence is not correct, please rewrite it.
Response 11: Line 264-265 “The absorbance and retention rate of SPI-7s-D3G were increased to 40.34%/37.95% and 17.33%/14.93%, respectively. Indicating that the thermal stability of anthocyanins decreased with the increase of pH, which may be caused by the denaturation of proteins during heating, resulting in structural changed that made protein bind more tightly to anthocyanins [21]. ” has been changed to “The absorbance and retention rate of SPI-7s-D3G were increased to 40.34%/37.95% and 17.33%/14.93%, respectively. These results indicated that SPI-7s improve the stability of D3G more efficiently in acidic condition than neutral condition.”
Point 12: lines 301-302 - the sentence is not correct and needs to be rewritten.
Response 12: Lines 301-302 “Above all, the interactions between protein and anthocyanins were different because of their different structures.” has been changed to “Thus, the interaction force between protein and anthocyanins depends on their structural characteristics.”
Point 13: lines 417-420 - the sentence is not clear, what do the Authors mean by: "position of hydroxyl and methoxy group in glycine"? - please explain.
Response 13: Line 417-420 “This may be due to the substitution pattern of anthocyanins, the number and position of hydroxyl and methoxy groups in glycine, which affected their chemical behaviors on pigment molecule, thus making protein and anthocyanins compound with different forces [42].” has been changed to “This may be due to the different structure of anthocyanins which affected their chemical behaviors on pigment molecule, thus making protein and anthocyanins compound with different forces [43].”
Point 14: lines 441-442 - the style of the sentence should be corrected.
Response 14: Line 441-442 “The stable system of soybean protein isolate-7s-delphinidin-3-O-glucoside (SPI-7s-D3G) was optimized using Box-Behnken design at pH 2.8 and pH 6.8 and the co-pigmentation rate were 43.62%/40.70%, respectively.” has been changed to “The stable system of SPI-7s-D3G (pH 2.8/pH 6.8) was optimized using Box-Behnken design, and the co-pigmentation rate of SPI-7s-D3G were increased to 42.62%/40.70%, respectively. ”
Point 15: lines 456-458 - style of the sentence is not correct (benefits on human body?)
Response 15: Line 456-458 “Considering the benefits of D3G and SPI-7s on human body, and SPI-7s can also improve the stability of anthocyanin, the stabilized SPI-7s-D3G will have potentially good application value in functional food.” has been changed to “Considering the potential health benefits of D3G and SPI-7s on human body, and SPI-7s can also improve the stability of anthocyanin, the stabilized SPI-7s-D3G will have potentially good application value in functional food.”
Point 16: line 105 - please explain what do the Authors mean by: "of aqueous"
Response 16: Line 105 “The eluents used consist of methyl alcohol (A) and 5% formic acid solution of aqueous (B).” has been changed to “ The eluents used consist of methyl alcohol (A) and 5% formic acid-water solution (B). ”

Reviewer 2 Report
As far as I am concerned the manuscript is well written. The subject area of research is significant in knowledge development. The introduction is interesting and correct. The results of the research were presented very well. The discussion of the results is good.
Red entries are misinforming, probably a remnant of language proofreading. This should be removed before being sent for review.
l.77 It would be necessary to explain once again what SPI is.
l. 112-158 Arrangement of patterns is not uniform. Pattern 3 has a description underneath it and should be on the same line. Pattern 1 and 4 is invisible, not separated from the text.
No spaces between chapters, text merges.
Author Response
Response to Reviewer 2 Comments
Point 1: 77 It would be necessary to explain once again what SPI is.
Response 1: Thanks for your correction. According to the opinions of the several reviewers. Soybean protein isolate-7s (SPI-7s) was explained in the abstract for the first time.
Point 2: 112-158 Arrangement of patterns is not uniform. Pattern 3 has a description underneath it and should be on the same line. Pattern 1 and 4 is invisible, not separated from the text.
Response 2: The format has been modified, please see the new manuscript.

Round 2
Reviewer 1 Report
I would like to thank the Authors for taking into account comments and recommendations of the Reviewer.
Below there are some additional comments, that should be reconsidered:
- Figures' titles: f.e.Figure 3. Absorbance value of heat treatment at 100 ℃ for 2 h. SPI-7s-D3G: The complex of soybean protein isolate-7s and delphinidin-3-O-glucoside; pH 2.8: The systems of pH 2.8; pH 6.8: The systems of pH 6.8.” Please explain, is it necessary to write: "pH 2.8: The systems of pH 2.8; pH 6.8: The systems of pH 6.8.”. In my opinion, it should be: "Figure 3. Absorbance value of heat treatment at 100 ℃ for 2 h. SPI-7s-D3G: The complex of soybean protein isolate-7s and delphinidin-3-O-glucoside" In the same way all the figures' titles should be corrected.
- Responce 11: "These results indicated that SPI-7s improve the stability of D3G more efficiently in acidic condition than neutral condition" - it should be: "improved"
- Responce 13: "This may be due to the different structure of anthocyanins which affected their chemical behaviors on pigment molecule, thus making protein and anthocyanins compound with different forces [43]" - what do the Authors mean by: thus making protein and anthocyanins compuond with different forces" - it is not clear - please correct the statement.
- Responce 14: "“The stable system of SPI-7s-D3G (pH 2.8/pH 6.8) was optimized using Box-Behnken design, and the copigmentation rate of SPI-7s-D3G were increased to 42.62%/40.70%, respectively" - it should be: " copigmentation rate of SPI-7s-D3G was increased to 42.62%/40.70%, respectively".
Author Response
Response to Reviewer 1 Comments
Point 1: Figures' titles: f.e.Figure 3. Absorbance value of heat treatment at 100 ℃ for 2 h. SPI-7s-D3G: The complex of soybean protein isolate-7s and delphinidin-3-O-glucoside; pH 2.8: The systems of pH 2.8; pH 6.8: The systems of pH 6.8.” Please explain, is it necessary to write: "pH 2.8: The systems of pH 2.8; pH 6.8: The systems of pH 6.8.”. In my opinion, it should be: "Figure 3. Absorbance value of heat treatment at 100 ℃ for 2 h. SPI-7s-D3G: The complex of soybean protein isolate-7s and delphinidin-3-O-glucoside" In the same way all the figures' titles should be corrected.
Response 1: Thanks for your correction.
“Figure 1. Absorbance value of sunlight treatment at room temperature. SPI-7s-D3G: The complex of soybean protein isolate-7s and delphinidin-3-O-glucoside; pH 2.8: The systems of pH 2.8; pH 6.8: The systems of pH 6.8.” has been changed to “Figure 1. Absorbance value of sunlight treatment at room temperature. SPI-7s-D3G: The complex of soybean protein isolate-7s and delphinidin-3-O-glucoside.” (Line 2551-252)
“Figure 2. Retention rate of sunlight treatment at room temperature. SPI-7s-D3G: The complex of soybean protein isolate-7s and delphinidin-3-O-glucoside; pH 2.8: The systems of pH 2.8; pH 6.8: The systems of pH 6.8.” has been changed to “Figure 2. Retention rate of sunlight treatment at room temperature. SPI-7s-D3G: The complex of soybean protein isolate-7s and delphinidin-3-O-glucoside.” (Line 254-255)
“Figure 3. Absorbance value of heat treatment at 100 ℃ for 2 h. SPI-7s-D3G: The complex of soybean protein isolate-7s and delphinidin-3-O-glucoside; pH 2.8: The systems of pH 2.8; pH 6.8: The systems of pH 6.8.” has been changed to “Figure 3. Absorbance value of heat treatment at 100 ℃ for 2 h. SPI-7s-D3G: The complex of soybean protein isolate-7s and delphinidin-3-O-glucoside.” (Line 307-308)
“Figure 4. Retention rate of heat treatment at 100 ℃ for 2 h. SPI-7s-D3G: The complex of soybean protein isolate-7s and delphinidin-3-O-glucoside; pH 2.8: The systems of pH 2.8; pH 6.8: The systems of pH 6.8.” has been changed to “Figure 4. Retention rate of heat treatment at 100 ℃ for 2 h. SPI-7s-D3G: The complex of soybean protein isolate-7s and delphinidin-3-O-glucoside.” (Line 310-311)
“Figure 5. Thermal degradation analysis at 100 ℃ for 100 min. SPI-7s-D3G: The complex of soybean protein isolate-7s and delphinidin-3-O-glucoside; pH 2.8: The systems of pH 2.8; pH 6.8: The systems of pH 6.8.” has been changed to “Figure 5. Thermal degradation analysis at 100 ℃ for 100 min. SPI-7s-D3G: The complex of soybean protein isolate-7s and delphinidin-3-O-glucoside.” (Line 313-314)
“Figure 6. Ultraviolet visible spectra of stable system. SPI-7s-D3G: The complex of soybean protein isolate-7s and delphinidin-3-O-glucoside; pH 2.8: The systems of pH 2.8; pH 6.8: The systems of pH 6.8.” has been changed to “Figure 6. Ultraviolet visible spectra of stable system. SPI-7s-D3G: The complex of soybean protein isolate-7s and delphinidin-3-O-glucoside.” (Line 330-331)
“Figure 7. Circular dichroism spectra and Fluorescence spectra of stable system. Circular dichroism spectra of pH 2.8 (A), Circular dichroism spectra of pH 6.8 (B), Fluorescence spectra of pH 2.8 (C), Fluorescence spectra of pH 6.8 (D). SPI-7s-D3G: The complex of soybean protein isolate-7s and delphinidin-3-O-glucoside; pH 2.8: The systems of pH 2.8; pH 6.8: The systems of pH 6.8; 0‒50: D3G concentration (μmol).” has been changed to “Figure 7. Circular dichroism spectra and Fluorescence spectra of stable system. Circular dichroism spectra of pH 2.8 (A), Circular dichroism spectra of pH 6.8 (B), Fluorescence spectra of pH 2.8 (C), Fluorescence spectra of pH 6.8 (D). SPI-7s-D3G: The complex of soybean protein isolate-7s and delphinidin-3-O-glucoside; 0‒50: D3G concentration (μmol).” (Line 35354)
“Figure 8. Stern-Volmer plots for SPI-7s-D3G (pH 2.8). SPI-7s-D3G: The complex of soybean protein isolate-7s and delphinidin-3-O-glucoside; pH 2.8: The systems of pH 2.8; pH 6.8: The systems of pH 6.8;289 K,308 K,318 K: The temperature at 298 K, 308 K or 318 K.” has been changed to “Figure 8. Stern-Volmer plots for SPI-7s-D3G (pH 2.8). SPI-7s-D3G: The complex of soybean protein isolate-7s and delphinidin-3-O-glucoside.” (Line 388-389)
“Figure 9. Stern-Volmer plots for SPI-7s-D3G (pH 6.8). SPI-7s-D3G: The complex of soybean protein isolate-7s and delphinidin-3-O-glucoside; pH 2.8: The systems of pH 2.8; pH 6.8: The systems of pH 6.8; 289 K,308 K,318 K: The temperature at 298 K, 308 K or 318 K.” has been changed to “Figure 9. Stern-Volmer plots for SPI-7s-D3G (pH 6.8). SPI-7s-D3G: The complex of soybean protein isolate-7s and delphinidin-3-O-glucoside.” (Line 391-392)
We also modified the notation of the table.
Point 2: Responce 11: "These results indicated that SPI-7s improve the stability of D3G more efficiently in acidic condition than neutral condition" - it should be: "improved"
Response 2: “These results indicated that SPI-7s improve the stability of D3G more efficiently in acidic condition than neutral condition. ” has been changed to “These results indicated that SPI-7s improved the stability of D3G more efficiently in acidic condition than neutral condition.” (Line 260-261)
Point 3: Responce 13: "This may be due to the different structure of anthocyanins which affected their chemical behaviors on pigment molecule, thus making protein and anthocyanins compound with different forces [43]" - what do the Authors mean by: thus making protein and anthocyanins compuond with different forces" - it is not clear - please correct the statement.
Response 3: “This may be due to the different structure of anthocyanins which affected their chemical behaviors on pigment molecule, thus making protein and anthocyanins compound with different forces [43]” has been changed to “The different interactions between anthocyanins and protein were attributed to the structural diversity of anthocyanins.” (Line 406-408)
Point 4: Responce 14: "“The stable system of SPI-7s-D3G (pH 2.8/pH 6.8) was optimized using Box-Behnken design, and the copigmentation rate of SPI-7s-D3G were increased to 42.62%/40.70%, respectively" - it should be: " copigmentation rate of SPI-7s-D3G was increased to 42.62%/40.70%, respectively".
Response 4: “The stable system of SPI-7s-D3G (pH 2.8/pH 6.8) was optimized using Box-Behnken design, and the co-pigmentation rate of SPI-7s-D3G were increased to 42.62%/40.70%, respectively.” has been changed to “The stable system of SPI-7s-D3G (pH 2.8/pH 6.8) was optimized using Box-Behnken design, and the co-pigmentation rate of SPI-7s-D3G was increased to 42.62%/40.70%, respectively.” (Line 423-425)

This manuscript is a resubmission of an earlier submission. The following is a list of the peer review reports and author responses from that submission.
Round 1
Reviewer 1 Report
The article titled: "Effect of soybean protein isolate-7s on delphinidin-3-O-glucoside from purple corn stability and their interactional characterization" needs major revision. First of all, it needs extensive editing of English language and style. In its present form, the article has serious flaws.
Comments and recommendations of Reviewer are listed below:
- lines 18, 19 - abbreviations: SPI and D3G should be explained as they appear for the first time in the text.
- line 26 - the style of the sentence should be corrected : binding - binding system.
- line 33 - what do the Authors mean by: human health effects.
- lines 48-51 - the sentence is unclear and should be rearranged.
- lines 51-52 - the change in protein structure is not a prove for improved anthocyanins stability - please explain.
- line 54 - it should be explained when the abbreviation appears for the first time in the text.
- line 61 - the sentence is unclear and should be rearranged.
- line 65 - "potentially" should be added, as it was not studied and presented in the article.
- lines 82-83 - it is a result and should be moved to "Results" section.
- line 85 - it should be: "anthocyanins" instead of: "antnocyanins"
- line 89 - Do the Authors mean: "concentrated" - please correct the sentence.
- line 101 - Do the Authors mean: :chromatogram" as it is: "is" in the sentence.
- line 106 - the sentence is unclear - it means that the Box-Behnken design was solubilized... - please rewrite the sentence.
- lines 118-120 - the same tense should be used throughout the whole article - please correct and decide what tense should be used, do not mix the tenses.
- line 123 - the sentence is incorrect - please rearrange it.
- line 141 - it should be: "were shown"
- line 144 b- it should be: "t is the time of heating"
- line 152 - what do the Authors mean by: "against"?
- line 167 mmol is not a concentration, do You mean mM?
- lines 178-179 - the sentence is unclear and should be rearranged.
- line 195 - please explain A, B, C.
- line 216 - do You mean - pretreatment takes a long time - it is not clear please correct the sentence.
- the sentence is incorrect - please rewrite it.
- line 255 - do You mean: "the highest" or "higher"?
- line 285 - do You mean: "interactions" as You used "were" in the sentence.
- line 292 - "reports have been reported" - the style of the sentence should be corrected.
- Figure 1, figure 3, figure 6 - why the abbreviation: "Abs" was used. The full name should be used as in other figures - please correct.
- line 317 - the sentence is unclear and should be rewritten.
- lines 329-330 - the sentence is unclear and should be rewritten.
- lines 333-334 - please use the same tense in the article, please unify it throughout the article, do not mix the tenses.
- line 339 -the sentence is gramtically incorrect - please rewrite it.
- lines 363-365 - the sentence is gramtically incorrect - please rewrite it.
- line 398 - it is not a reaction - please correct the sentence.
- lines 401, 412 - "shown" is gramatically not a correct form here - please correct.
- line 428 - "was" is missing in the sentence.
- line 436 - the sentence is incorrect.
- line 438 - what does it mean: "produced co-pigmentation"?
- line 448 - how do You know, that such complex could be widely used in functional foods, as You did not conduct such researches?
- Figure 1 - It should be:"chromatogram" instead of: "chromatograms".
Reviewer 2 Report
The Authors focused in this manuscript on spectroscopic analysis of developed system soybean protein isolate-7s on delphinidin-3-O-glucoside. Unfortunately, in my opinion, the manuscript is out of the scope of the Journal. The main analysis of the system was focused on spectroscopy, and the connection with food is very weak. I m recommending shifting this manuscript i.e. to Applied Sciences or other more focused journals.
Detailed comments were listed below:
Abstract
I did not see clearly the connections with foods. Please highlight it, as well as a short comment on why did you choose described system (i.e. see the first sentence in the introduction).
Introduction
- 39 743.158 ± 11.304 and 552.381 ± 9.1 mg - there is no need to present the numbers in this form (you can give general information). In the present form, it is more suitable for the discussion section. What is more some references are needed here
line 43 stabilization of what?
line 55 some references here will be beneficial
At the end of the introduction, typically "some introduction" for the next part of manuscript content is desirable. Just briefly describe what you did (see some previously published papers i.e. in Foods, etc.).
2.4 consider changing the head title i.e. "experimental design" (not obligated)
Results
Here, where the manuscript is focused on the application and usefulness of the BB method, the description of -1 0 1, etc. should be briefly described (i.e. under the table). As I understood, the manuscript showed the desirable benefits from statistical methods for quick "optimal" products preparation.
Why did you choose pH 2.8 and 6.8?
section 3 should be named results and discussion